

**A new approach for modeling suspended sediment: Evolutionary fuzzy approach**
Ozgur Kisi
Civil Engineering Department, Faculty of Engineering and Architecture, Canik Basari
University, Samsun, Turkey; E-mail: okisi@basari.edu.tr; Phone: +90 (362) 280 10 70





**Abstract:** This paper proposes the application of evolutionary fuzzy (EF) approach for prediction
of daily suspended sediment concentration (SSC). The EF was improved by the combination of
two methods, fuzzy logic and genetic algorithm. The accuracy of EF models is compared with
those of the artificial neural network (ANN) and adaptive neuro-fuzzy inference system with
fuzzy c-means clustering (ANFIS-FCM). The daily streamflow and suspended sediment data
collected from two stations on the Eel River in California, United States are used in the study.
Root mean square errors (RMSE), mean absolute errors (MAE) and determination coefficient
criteria are used for evaluating the accuracy of the models. The EF is found to be superior to the
ANN and ANFIS-FCM in SSC prediction. The relative RMSE and MAE differences between the
optimal EF and ANN models were found to be 13-50% and 15-65% for the upstream and
downstream stations, respectively. Comparison of the optimal EF, ANN and ANFIS-FCM
models in estimating peak and total suspended sediments revealed that the EF model provided
better accuracy than the ANN and ANFIS-FCM.
**Keywords:** Suspended sediment concentration; modelling; neural networks; fuzzy logic; genetic
algorithm.
**1. Introduction**
The sediment transport in rivers is vital for pollution, channel navigability, reservoir filling,
hydroelectric-equipment longevity and scientific interests. The assessment of the sediment
amount being transported by a river has a vital importance in hydraulic engineering due to its
importance in the design and management of water resources projects (Jain 2001; Kisi et al.
2006). The suspended sediment estimation is exceedingly difficult since it is closely related to
flow and their relationship mechanism is highly non-linear and they have complicated
interactions to each other (Sivakumar and Wallender 2005).





Artificial neural networks (ANN) have been successfully applied in water resources in the last
decades (Kisi and Shiri 2012; Kisi et al. 2012, 2013; Shiri et al. 2013). Recent investigations have
reported that ANNs may offer a promising alternative for suspended sediment estimation (Jain
2001; Tayfur 2002; Kisi 2005; Cigizoglu & Kisi 2006, 2008, 2010; Dogan et al. 2007; Rai and
Mathur 2008; Kisi et al. 2008, 2009; Cobaner et al. 2009; Jothiprakash and Garg 2009; Rajaee et
al. 2009; Talebizadeh et al. 2010; Melesse et al. 2011; Mustafa et al. 2012; Kisi and Aytek 2013;
Kitsikoudis et al. 2014). Jain (2001) compared single ANN approach with rating curve in
establishing sediment-discharge relationship and found that the ANN model performed better
than the rating curve. Tayfur (2002) used an ANN model for sheet sediment transport and
compared with physically-based models, whose transport capacity was based on one of the
dominant variables-flow velocity, shear stress, stream power, and unit stream power. He reported
that the ANN performed as well as, in some cases better than, the physically-based models. Kisi
(2005) used an ANN model for estimating suspended sediment and compared its results with
sediment rating curve (SRC) and multiple linear regression (MLR). He used daily streamflow and
suspended sediment data from two stations, Quebrada Blanca and Rio Valenciano, operated by
the US Geological Survey. Comparison results indicated that the ANN model performs better
than the regression and rating curve techniques in estimation of suspended sediment. Cigizoglu &
Kisi (2006) proposed some methods to improve ANN accuracy in suspended sediment
estimation. They used k-fold partitioning of the training data set and showed that similar or even
superior sediment estimation performances can be obtained with quite limited data provided that
the training data statistics of the subset are close to those of the testing data. Rai and Mathur
(2008) developed a back propagation feed-forward ANN model for the computation of event-
based temporal variation of sediment yield from the watersheds and compared with linear transfer
function model. Based on the comparison, the ANN based model resulted better agreement than



the linear transfer function model for the computation of runoff hydrographs and
sedimentographs for both the watersheds (W-2 watershed of Treynor catchment and W7 of
Goodwin Creek experimental watershed in USA). Kisi (2008) compared three different ANN
training algorithms in suspended sediment estimation by using the flow and sediment data from
the stations Quebrada Blanca and Rio Valenciano in USA. He indicated that the Levenberg–
Marquardt and conjugate gradient algorithms performed better than the gradient descent in
suspended sediment estimation. He also reported that the gradient descent algorithm took an
unusually high number of iterations and time taken by the other two algorithms for training of the
network. Jothiprakash and Garg (2009) used ANN model for estimating the volume of sediment
retained in a reservoir and they found that the ANN model estimated the sediment volume with
better accuracy and less effort as compared to conventional regression analysis. Rajaee et al.
(2009) compared ANN with MLR and SRC models in daily simulation of suspended sediment
concentration (SSC) using daily river discharge and SSC data from the Little Black River and
Salt River stations in the USA. They indicated that the ANN model was more accurate than the
MLR and SRC models in predicting SSC. Talebizadeh et al. (2010) made uncertainty analysis in
sediment load estimation by using ANN and SWAT model. Melesse et al. (2011) used ANNs
with an error back-propagation algorithm to predict the suspended sediment load from
Mississippi, Missouri, and Rio Grande major rivers in USA. They evaluated different input
combinations and compared the results with MLR, multiple nonlinear regression and
autoregressive integrated moving average (ARIMA). Kisi and Aytek (2013) proposed explicit
neural network (ENN) formulation for modeling daily suspended sediment-discharge relationship
and compared with two different SRCs, MLR and nonlinear regression (NLR). They used daily
streamflow and suspended sediment data from two stations on Tongue River in Montana, USA.
The comparison results revealed that the ENN model performs better than the conventional SRC,



MLR and NLR. Kitsikoudis et al. (2014) employed ANN and ANFIS for prediction of bed load
transport rates in gravel-bed steep mountainous streams and rivers in Idaho (USA). They
compared ANN and ANFIS results with those of the symbolic regression (SR) based on genetic
programming (GP) and widely applied bed load formulas. The ANN and ANFIS models
performed equally well, better than SR and bed load formulas. Mustafa et al. (2012) compared
different ANN training algorithms (gradient descent, gradient descent with momentum, scaled
conjugate gradient, and Levenberg-Marquardt) in prediction of the suspended sediment discharge
of Pari River at Silibin in Peninsular Malaysia and they found that Levenberg-Marquardt (LM)
was faster and more powerful than the other algorithms. In the present study, also, the LM is used
for training ANN models.
Fuzzy logic has also been successfully employed for suspended sediment estimation during
recent years (Tayfur et al. 2003; Kisi 2009; Lohani et al. 2007; Kisi et al. 2006, 2008, 2009;
Mirbagheri et al. 2010; Wieprecht et al. 2013; Kitsikoudis et al. 2014; Roushangar 2014). Tayfur
et al. (2003) applied fuzzy logic approach for modeling runoff-induced sediment transport from
bale soil surfaces and obtained satisfactory results. They compared the fuzzy model with those of
the physics-based models in predicting the mean sediment loads from experimental runs. The
results indicated that the fuzzy model performed better than the physically-based model under
very high rainfall intensities over different slopes and over very steep slopes under different
rainfall intensities. Kisi et al. (2006) applied fuzzy logic approach to 5-year period of continuous
streamflow and sediment concentration data of Quebrada Blanca Station operated by the United
States Geological Survey. They indicated that fuzzy rule based models performs better than the
SRC models in prediction of daily suspended sediment concentration. Lohani et al. (2007) used
adaptive neuro-fuzzy inference system (ANFIS) for developing stage-discharge-sediment
concentration relationships by using data from two gauging sites in the Narmada basin in India.



Comparison results revealed that the ANFIS model significantly improved the magnitude of
prediction accuracy and could be successfully applied for sediment concentration prediction. Kisi
et al. (2008) modeled daily suspended sediment estimation by ANFIS models and compared with
radial basis neural network (RBNN), feed-forward neural network (FFNN), generalized
regression neural network (GRNN), MLR and SRC. They used daily streamflow and suspended
sediment data of four stations in the Black Sea region of Turkey. They reported that the ANFIS
model, in general, gave better estimates than the other models. Kisi et al. (2009) investigated the
accuracy of an ANFIS computing technique in monthly suspended sediment estimation. They
used monthly streamflow and suspended sediment data from two stations, Kuylus and Salur
Koprusu, in Kizilirmak Basin in Turkey. They obtained better estimates than the conventional
SRC. Mirbagheri et al. (2010) used ANFIS method for SSC prediction by using daily data from
the Rio Rosario gauging station, Puerto Rico, USA. They found that proposed ANFIS model was
able to improve on the RMSE value of the SRC method by about 44.32%. Roushangar (2014)
applied ANFIS method for modeling of total bed material load through developing the accuracy
level of the predictions of traditional models. They used data of Qotur River (Northwestern Iran).
The comparison results indicated that the ANFIS models performed better than the sediment
transport formulas in modeling total bed material load transport rate. They also found that the
models based on stream power approach (used by Bagnold and Engelund-Hansen) were more
reliable than those based on shear stress approach (used by Laursen) in estimating sediment
transport rate. It is apparent from the literature that no work has reported the use of fuzzy genetic
approach for modeling SSC.
This study investigates the applicability of fuzzy genetic approach for predicting daily SSC. The
evolutionary fuzzy (EF) models are compared with those of the ANN and ANFIS with fuzzy c-
means clustering (ANFIS-FCM) models. To the best knowledge of the author, this is the first



study that compares the accuracy of EF model with those of the ANN and ANFIS-FCM models
in suspended sediment modeling.
**2. Methods and materials**
*2.1. Fuzzy logic approach*
Fuzzy logic is firstly introduced by Zadeh (1965) and used different scientific researches. The
fuzzy concepts and algorithms can be found in many related textbooks (Kosko 1993; Ross 1995).
The fuzzy logic (FL) theory has a mechanism for representing linguistic constructs such as
"high", "low", "medium", "few" etc. The FL has an inference system that enables human
reasoning capabilities while the conventional binary set theory defines crisp events. The FL
theory is based upon the notion of relative graded membership degree between 0 and 1.0. The
fuzzy sets have ability to model indistinct or ambiguous data, often faced in real life (Sivanandam
et al. 2007).
As seen from Figure 1, a typical fuzzy inference system is a rule-based system and composed of
three conceptual components. These are; 1) a rule base comprising fuzzy IF-THEN linguistic
rules relates the membership functions (MFs) of the input variables to the outputs' MFs; 2) a
database consisting membership functions used in fuzzy linguistic rules; 3) an inference
mechanism that incorporate these rules to relate a set of outputs to a set of inputs and to obtain a
reasonable output. In the fuzzification, input and/or output data are considered as having
ambiguous characteristics and therefore, they are divided into subsets defined by linguistic terms
(e.g., small, big) and membership degrees are determined. In the defuzzification, a crisp
numerical value is computed from the fuzzy linguistic outputs obtained from the inference
mechanism (Nayak et al. 2005). The part between IF and THEN is called antecedent, while the
part after THEN is referred to consequent.





Let assume that the input and output variables are partitioned into subsets with Gaussian fuzzy
MFs. If there are three input variables comprising two membership functions in the antecedent
part, there should be $2^3$ rules in the fuzzy rule base. Increasing number of subsets may results in
better accuracy. In this case, however, the rule base gets larger and its construction will be
difficult to construct (Şen 1998). Assume that we have two inputs with two fuzzy subsets or MFs
labeled as "weak" and "strong" and one output then there should be four rules as follows:

7         $R_1$: IF $x_1$ is weak and $x_2$ is weak THEN $y_1$

8         $R_2$: IF $x_1$ is weak and $x_2$ is strong THEN $y_2$

9         $R_3$: IF $x_1$ is high and $x_2$ is weak THEN $y_3$

10        $R_4$: IF $x_1$ is strong and $x_2$ is strong THEN $y_4$

where $x_1$ and $x_2$ are input1 and input2 and $y_1$, $y_2$, $y_3$ and $y_4$ are constant or linear equations.
In each fuzzy model used in the present study, membership degrees, $w_n$, for $x_1$ and $x_2$ are
computed to be assigned to the corresponding output $y_n$ for each triggered rule. Thus, a single
weighted output, $y$, is computed by weighting average of the outputs obtained from four rules as:

$$y = \frac{\sum_{n=1}^{4} w_n \cdot y_n}{\sum_{n=1}^{4} w_n} \qquad (1)$$

The output values, $y$, can be simply calculated from Eq. (1) for any input combination after
setting up the rule base (Şen 1998).
*2.2. Genetic algorithm*
Holland (1975) explained in his book how to apply the principles of natural evolution to
optimization problems and built the first genetic algorithms (GAs). In the last decades, GAs have
been used as a powerful means for solving search and optimization problems (Sivanandam and




Deepa, 2008). The main idea in GAs is to simulate the natural evolution mechanisms of
chromosomes, including the rudimentary elements of natural genetics for example reproduction,
crossover, and mutation.
Three core steps are included in a typical form of a GA (Preis and Ostfeld, 2008):
i.   Generation of initial population: GA produces a set of strings (or population), with each
string (chromosome) containing a set of parameter values to be optimized.
ii.   Strings fitness calculation: GA assesses the fitness of each string (i.e., the objective
function value).
iii.  Production of new generation: The next generation is produced by performing selection,
crossover and mutation. Selection is used to choose chromosomes from the recent
population for reproduction with respect to fitness values.
One of the main reproduction operator employed is bit-string crossover (Figure 2). In this
operator, two strings are used as parents and new individuals are generated by swapping a sub-
sequence between the two strings. The other main operator is bit-flipping mutation (Figure 3). In
this operator, a single bit in the string is flipped to constitute a new offspring string. All operators
in GA are delimited to manipulate the string in a parallel manner to the structural interpretation of
genes. For instance, two genes in the same location on two strings may be exchanged between
parents, but not merged based on their values. Individuals are usually selected to be parents
probabilistically with respect to their fitness values, and the offspring that are formed replace the
parents (Sivanandam and Deepa, 2008).
GA is a powerful method with regard to search the optimum solution to complex
problems such as the choice of the MFs where it is hard or almost impossible to test for
optimality (Ahmed and Sarma, 2005).
The main differences between GAs and conventional optimization methods are:



• The parameter sets are coded in GAs, not the parameters.
• Local optimum is explored from a population in GAs, not a single point.
• The objective function information is used in GAs, not adjutant knowledge (e.g.
derivatives).
• Probabilistic evolution rule is used in GAs, not deterministic rules (Goldberg, 1989).
The GA explores for the best potential solutions of a problem from existing solution sets.
The problem is converted to binary form and the solutions are allowed to crossover and mate
with a specified criterion to yield the optimal. The basics of the GA can be obtained from Wang
(1991), Ahmed and Sarma (2005).
*2.3. Evolutionary Fuzzy Approach*
In this study, the EF was developed by the combination of two methods, fuzzy logic and genetic
algorithm. The optimal parameters (e.g. antecedent and consequent parameters) of the fuzzy
models were obtained by using genetic algorithms. Figure 4 demonstrates the flowchart of a
fuzzy genetic model. Genetic algorithm optimization is done by minimizing the error (objective
function) between model estimates and measured values. In this study, mean square error was
used as objective function in genetic algorithm. The MSE can be expressed as

$$MSE = \frac{1}{N} \sum_{i=1}^{N} \left( yi_{observed} - yi_{model} \right)^2 \tag{2}$$

where $N$ is the number of training data. Here, the objective function given in Eq. 2 was
minimized by adjusting the MF parameters of the input and outputs. The optimization of the MFs
is a complex problem for the supervised learning scheme. Genetic algorithm, however, has a non-
supervised learning scheme and can be successfully applied to solve this problem (Goldberg
1989, Ozger 2009).



*2.4. Case Study*
The daily streamflow and SSC data from two stations, upstream station near Dos Rios (station
No: 11147000) and the downstream station at Scotia (station No: 11472150), on the Eel River in
California were used in the present study. The stations are operated by the US Geological Survey
(USGS). The drainage areas of the upstream and downstream stations respectively are 1368 km$^2$
and 8063 km$^2$. Daily data were downloaded from the web server of the USGS
(http://webserver.cr.usgs.gov/sediment). In the both stations, the data from October 01, 1966 to
September 30, 1971 were used for training, the data from October 01, 1971 to September 30,
1974 were used for validation and the data from October 01, 1974 to September 30, 1977 were
used for models' testing. Streamflow and suspended sediment data of upstream and downstream
stations are shown in Figures 5-6. In California rivers (e.g. Eel River), the geologic, climatic,
physiographic, and land-use conditions are highly variable (Tramblay *et al.* 2010). An
extraordinary flood was occurred on the Eel River near Scotia, California (downstream station,
11477000) in 1964. This is one of the most widespread and destructive floods in the history of
the West Coast (Waananen *et al.* 1971). The Eel River is the most exceptional flood-producing
river in the United States (O'Connor and Costa 2004). On December 23, 1964, the Eel River at
Scotia, California, peaked up at a stage of 72 ft and a discharge, designated by a rating curve
extension, of 752,000 ft$^3$/s. For measuring peak discharges above a threshold at this site, surface
velocities measured by optical current meter are used. The Eel River is may be the only site in the
US, where optical current meters are routinely used for high-flow discharge measurements (Costa
and Jarrett 2008). Groundwater recharge, recreation, and industrial, agricultural and municipal
water supply were supplied from the river (Brown and Ritter, 1971). The Eel River system is
among the most dynamic in California due to the region's unsteady geology and the effect of





major Pacific storms. The discharge is highly variable in this river; average flows in January and
February are over 100 times greater than in August and September (USGS 2013). The Eel River
also conveys the highest suspended sediment load of any river of its size in the United States, in
part as a result of the frequent landslides in the region. Unlike most areas, suspended sediment
discharge per unit area in the river increases with catchment size (Brown and Ritter, 1971; Janda
and Nolan, 1979). As a result of ongoing uplift, main channels are generally more deeply incised
than their tributaries, and so streamside landslides, which are major sources of sediment, are
mainly plentiful along main channels. Parent material is mostly soft and friable, and therefore,
bed particles quickly break down into smaller sizes (Knott, 1971). Accordingly, suspended-
sediment load growths downstream at the expense of bedload (Brown and Ritter, 1971; Lisle

11    2013).

Statistical parameters of daily streamflow and SSC data are shown in Table 1 for the upstream
and downstream stations. In this table, $S_x$, $C_v$, $C_{sx}$, $x_{mean}$, $x_{max}$ and $x_{min}$ are the standard deviation,
variation coefficient, skewness coefficient, mean, maximum and minimum, respectively. From
the table it is clear that the flow and SSC data have a considerably high skewed distribution
(range 8.05-19.5 for the upstream station and range 7.11-14.5 for the downstream station). The
validation and test data indicate much more skewed distribution than those of the training data for
the both stations. The maximum-mean ratios ($x_{max}/x_{mean}$) for SSC series are also quite high
especially for the validation and test data (244-154 and 135-156 for the upstream and
downstream, respectively). It is evident from these statistics that the discharge-sediment
phenomenon has a highly complex behavior.
**3. Results and discussion**





Different EF models were tried in terms of number of membership functions and generations.
The EF models were compared with ANN and ANFIS-FCM models. Three different program
codes, including fuzzy logic, genetic algorithm and neural network toolboxes, were written in
MATLAB language for the simulations of EF, ANN and ANFIS-FCM models.
Root mean square errors (RMSE), mean absolute errors (MAE) and determination
coefficient ($R^2$) were used for evaluation of the applied models. The RMSE, MAE and $R^2$
statistics are expressed as

$$RMSE = \sqrt{\frac{1}{N}\sum_{i=1}^{N}(SSCi_{observed} - SSCi_{predicted})^2} \qquad (2)$$

$$MAE = \frac{1}{N}\sum_{i=1}^{N}\left|SSCi_{observed} - SSCi_{predicted}\right| \qquad (3)$$

$$R^2 = \left( \frac{\sum_{i=1}^{N}(SSCi_{observed} - \overline{SSC}_{observed})(SSCi_{predicted} - \overline{SSC}_{predicted})}{\sqrt{\sum_{i=1}^{N}\left(SSCi_{observed} - \overline{SSC}_{observed}\right)^2 \sum_{i=1}^{N}\left(SSCi_{predicted} - \overline{SSC}_{predicted}\right)^2}} \right)^2$$
$$(4)$$

in which $N$ is the number of data, $SSC$ is the suspended sediment concentration, $\overline{SSC}$ is mean of
the $SSC$.
Various input combinations including previous streamflows (Table 2) were tried to
estimate suspended sediment concentrations of the upstream station. In this table, Qt and Qt-1
indicate the discharge at current and one previous days, respectively. Input combinations were
determined according to the correlation analysis given in Table 2 and following the related
literature (Jain, 2001; Kisi, 2005). For each input combination, optimum parameters of the EF,
ANN and ANFIS-FCM models were obtained by minimizing the objective function (MSE error
between calculated and observed SSC values) in validation period. The training and validation



results of the EF, ANN and ANFIS-FCM models are shown in Table 3 for the upstream station.
The computing times required for the applied models are also compared in this table. In this
table, the EF(3,gauss,5000) model has the 3 Gaussian MFs for the inputs, Qt, Qt-1, Qt-2 and Qt-3
and 5000 generations. ANN(4,2,1) indicates an ANN model comprising 4 input, 2 hidden and 1
output nodes. ANFIS-FCM(5) model has 5 cluster or 5 Gaussian MFs for each input. In all ANN
models, the logarithm sigmoid activation function commonly used in the literature was used for
the hidden and output nodes. It is evident from Table 3 that the ANFIS-FCM models generally
perform better than the EF and ANN models in the validation period. The ANFIS-FCM models
require less computing time for calibration than the other models. The EF models has the most
computing time in calibration (training). Table 5 compares the accuracy of the applied models in
the test period. It is obvious from the table that all the EF models generally have better accuracy
than the ANN and ANFIS-FCM models. The relative RMSE and MAE differences between the
optimal EF (input combination iii) and ANN (input combination iv) models are 13% and 50%,
respectively. Figure 7 illustrates the scatterplots of the optimal EF, ANN and ANFIS-FCM
models in the test period for the upstream station. The $R^2$ value of ANN seems to be slightly
higher than the EF model. However, the *a* and *b* fit line equation coefficients of the EF model
(assume that the equation is y=*a*x+*b*) respectively closer to the 1 and 0 than those of the ANN
model. Figure 8 demonstrates the log-scaled scatterplots of the optimal models in test period. The
peak SSC estimates of the ANN model seem to be closer to the exact line than those of the EF
and ANFIS-FCM. However, the EF and ANFIS-FCM models seem to be better the ANN model
in low sediment estimation. It should be noted that the distribution of the EF and ANFIS-FCM
models' estimates are similar to each other. Table 6 reports the comparison of the models' SSC
peak-estimates. It is evident from the table that the EF model gives better estimates of peak SSC
values than the ANN and ANFIS-FCM models. The ANFIS-FCM is the second best in



estimating peak SSC. The EF and ANN models respectively estimated the observed total
sediment load, 20,289,369 ton, as 30,308,073 ton and 31,820,879 ton with overestimations of
49.4% and 56.8% while the ANFIS-FCM model resulted in 9,514,219 ton with an
underestimation of 53.1%. The EF model seems to be slightly better than the other models in
estimating total sediment load.

6        Same input combinations were used to estimate SSC values for downstream station. The

training and validation results of the EF, ANN and ANFIS-FCM models are given in Table 4.
The architectures of the EF, ANN and ANFIS-FCM models are also provided in the first column
of this table. From the table, it is clear that the ANN models perform better than the EF and
ANFIS-FCM in validation period. Here also ANFIS-FCM models require less computing time
for calibration than the other models while the EF models has the most computing time in
training. Comparison of Table 3 and 4 clearly reveals that the models' accuracies are better in
upstream station than the downstream. The reason of this may be the fact that the downstream has
much larger drainage area than the upstream and the SSC in downstream may be much more
affected by perturbations (urbanization, land-use change, slope failures, forest fires, earthquakes,
etc.). Table 5 compares the test accuracy of the models with respect to RMSE, MAE and $R^2$
values. Here, also the EF models perform better than the ANN and ANFIS-FCM models. The
relative RMSE and MAE differences between the optimal EF (input combination iii) and ANN
(input combination iv) models are 15% and 65%, respectively. The observed and estimated SSC
values by the optimal EF, ANN and ANFIS-FCM models in the test period are shown in Figure 9
for the upstream station. It is evident from the fit line equations and $R^2$ values that the EF
estimates are closer to the exact line than those of the ANN and ANFIS-FCM models. The log-
scaled scatterplots of the optimal models are compared in Figure 10 for the test period. The EF
model seems to have better accuracy in estimating average and low SSC values than the ANN





and ANFIS-FCM models. The comparison of the optimal models' peak SSC estimates are made
in Table 6 for the downstream station. The superior accuracy of the EF model to the ANN and
ANFIS-FCM models is clearly seen from this table. The EF model estimated total sediment load
as 6,252,171,710 ton instead of observed value of 5,051,074,055 ton, with an overestimation of
24% while the ANN and ANFIS-FCM models resulted in 6,422,653,088 ton and 6,464,368,673
ton with overestimations of 27% and 28%, respectively. The results indicate that all applied
models generally provided overestimations for the peak and total SSC in both stations. The main
reason of this may be the differences between training, validation and testing datasets. It is clear
from Table 1 that the validation data set has much higher SSC values (661,000 ton for the
upstream and 6,230,000 ton for the downstream) than those of the test data set (86,500 ton for the
upstream 2,870,000 ton for the downstream) in both upstream and downstream stations. The
significantly high streamflow and suspended sediment values are also clearly seen from Figures
5-6. It is clear from the figures that the high floods (e.g. 1590 m$^3$/s and 9170 m$^3$/s in 16 Jan 1974
for the upstream and downstream, respectively) occurred in validation period causes high SSC
values (e.g. 661,000 mg/l and 6,230,000 mg/l in 16 Jan 1974 for the upstream and downstream,
respectively). The great changeability of Eel River in space and time can be obviously seen from
the figures. The optimal models were obtained according to their minimum MSE errors in the
validation period. Therefore, the high SSC values in this period lead models to give
overestimations in the test period. The difference between the validation and test data sets may be
due to the fact that extreme SSCs in Californian Rivers show a great changeability in space and
time, and are interrelated with some physiographic features at the station locality scale (Tramblay
*et al.* 2010). O'Connor and Costa (2004) reported that the Eel River is the most exceptional
flood-producing river in the United States.





In overall, the EF models seem to be more adequate than the ANN and ANFIS-FCM for
estimating SSC. The main disadvantage of the ANN is its black-box structure. The other
disadvantage is it uses backpropagation (BP) methodology for adjusting the weights and it is very
easy for the training process to get trapped in a local minimum (Kumar *et al*., 2002; Sudheer *et*
*al*., 2003). By combining ANN and fuzzy (neuro-fuzzy), the individual strengths of each
approach can be employed in a synergistic way for the building effective and powerful intelligent
systems. Neuro-fuzzy (e.g., ANFIS) methods have the ability to get the benefits of both these
fields in a single system. The drawback of fuzzy system design (getting a set of fuzzy if-then
rules) is amended by ANFIS system where the learning ability of an ANN is used, automatic
fuzzy if-then rules are generated and parameters are optimized (Jang, 1993; Nayak et al., 2004).
The EF and ANFIS models use transparent, linguistic representation of a fuzzy system and
provide set of rules on which the model is based. This provides further insight into the modeled
process (Sayed *et al.*, 2003). In ANFIS, however, gradient descent algorithm is used for the
determination of membership functions (MFs). The main disadvantage of this algorithm is that it
uses BP methodology for amending the weights and it is very easy for the calibration process to
get trapped in a local minimum (Kumar *et al*., 2002; Sudheer *et al*., 2003). The main advantage
of EF compared to ANFIS is that it uses genetic algorithm. Genetic algorithm combines
stochastic and directed search elements and they offer global optimum without being trapped in
local optima (Mantoglou *et al.*, 2004; Karterakis *et al.*, 2007). The main disadvantage of the EF is
that it requires long time for calibration.
**6. Conclusions**
In this paper, the applicability of fuzzy genetic approach for prediction of daily suspended
sediment concentration was investigated. The EF models' accuracy is compared with those of the





artificial neural networks and adaptive neuro-fuzzy inference system with fuzzy c-means
clustering. The daily streamflow and SSC data from two stations on the Eel River in California
were used in the applications. Various input combinations consisting previous streamflows were
used as inputs to the EF, ANN and ANFIS-FCM models in order to estimate SSC of the upstream
and downstream stations. For the both stations, the best EF and ANFIS-FCM models were
obtained for the third input combination composed of current and two previous streamflow data
while the ANN model gave the best accuracy for the inputs, Qt, Qt-1, Qt-2 and Qt-3 (fourth input
combination). The comparison of the EF, ANN and ANFIS-FCM models showed that the EF
models performed better than the ANN and ANFIS-FCM. The optimal EF, ANN and ANFIS-
FCM models were also compared with each other in estimating peak and total suspended
sediments and results indicated that the EF model generally provided better accuracy than the
ANN and ANFIS-FCM. The results suggest that the EF can be successfully used for developing
streamflow-sediment relationship in the rivers where the geologic, climatic, physiographic, and
land-use conditions are highly variable.
**Acknowledgements**
This study was supported by The Turkish Academy of Sciences (TUBA). The author would like
to thank TUBA for their support of this study. The data used in this study were downloaded from
the web server of the USGS. The authors wish to thank the staff of the USGS who are associated
with data observation, processing, and management of USGS web sites.

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





**Figure Captions**
**Figure 1** A typical fuzzy inference system
**Figure 2** Bit-string crossover of parents (i) and (ii) to form offspring (iii) and (iv)
**Figure 3** Bit-flipping mutation of parents (ii) to form offspring (ii)
**Figure 4** The flowchart of the fuzzy genetic model (Kisi and Tombul, 2013)
**Figure 5** Streamflow and suspended sediment data of upstream station.
**Figure 6** Streamflow and suspended sediment data of downstream station.
**Figure 7** Scatterplots of the observed and estimated SSC by EF, ANN and ANFIS-FCM -
Upstream station.
**Figure 8** Scatterplots of the observed and estimated SSC by EF, ANN and ANFIS-FCM
(logarithm scaled) - Upstream station.
**Figure 9** Scatterplots of the observed and estimated SSC by EF, ANN and ANFIS-FCM -
Downstream station.
**Figure 10** Scatterplots of the observed and estimated SSC by EF, ANN and ANFIS-FCM
(logarithm scaled) - Downstream station.





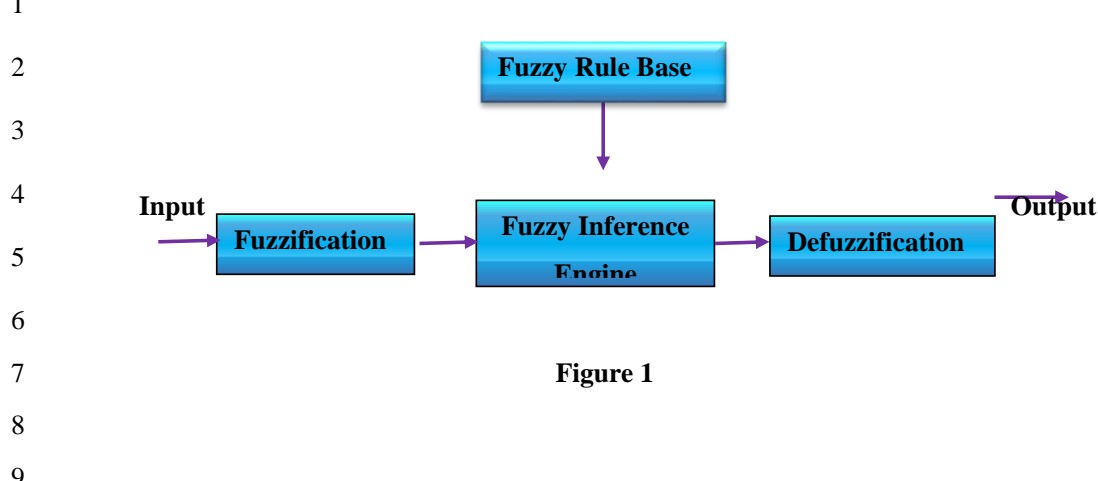

Figure 1



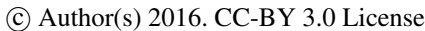

**Figure 2**



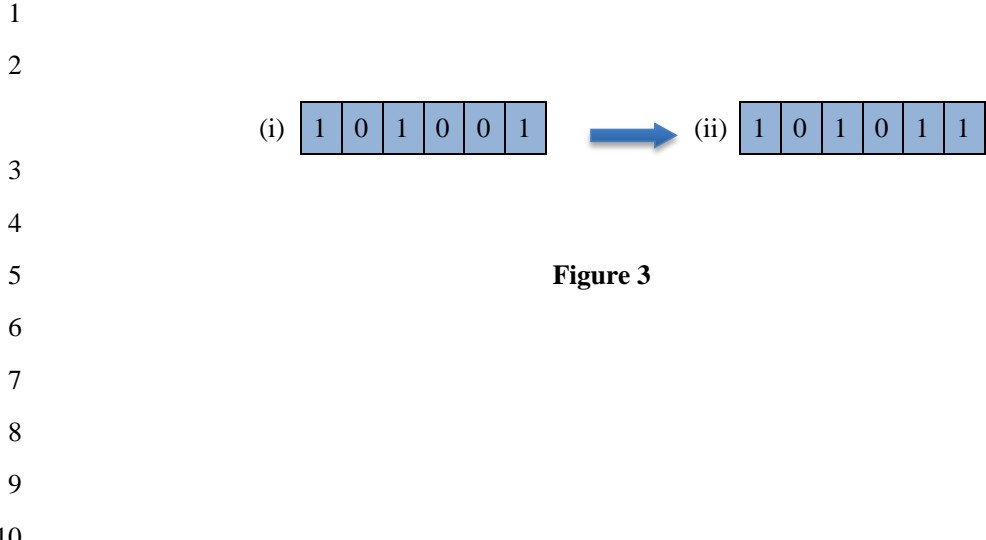

5                          **Figure 3**








**Figure 4**



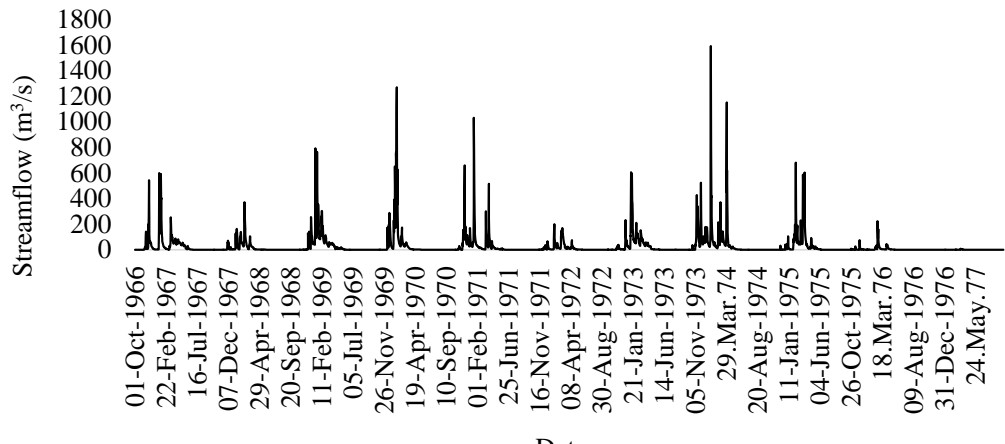

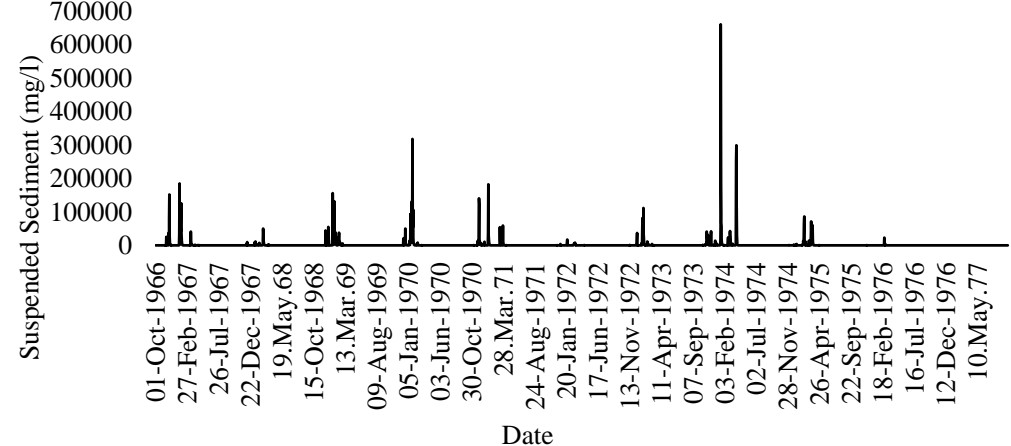

6 **Figure 5**



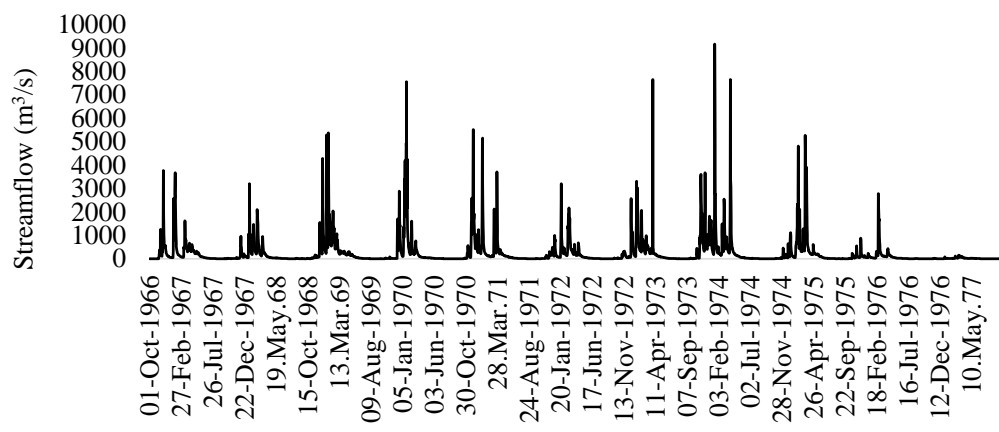

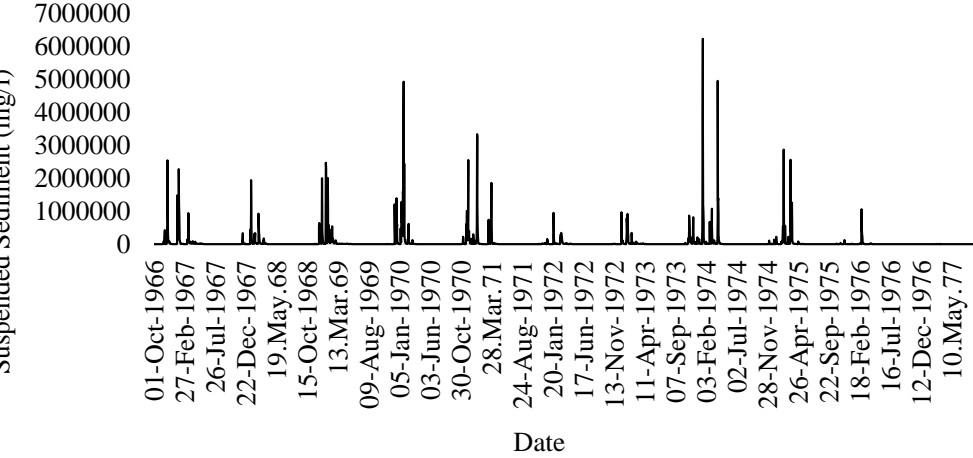

6 **Figure 6**

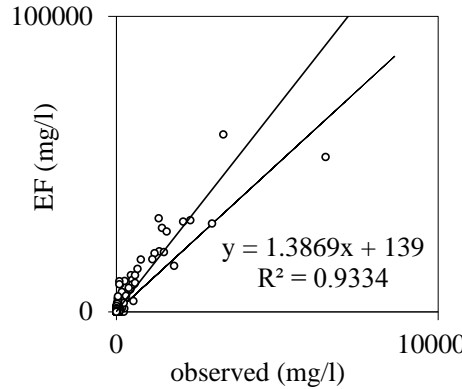

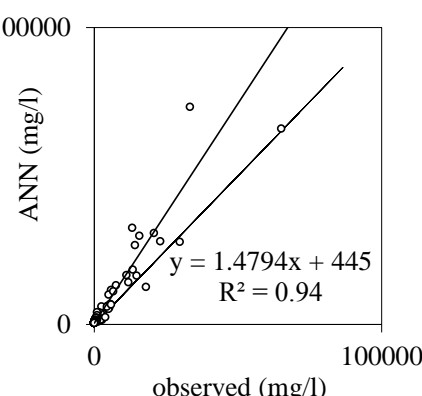

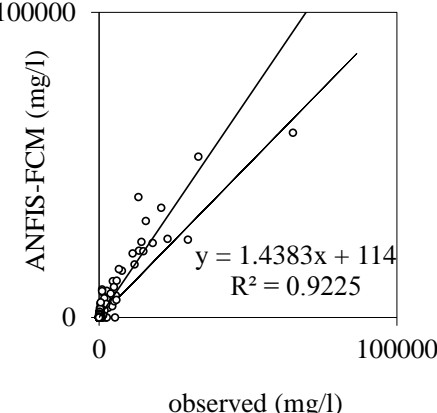

**Figure 7**



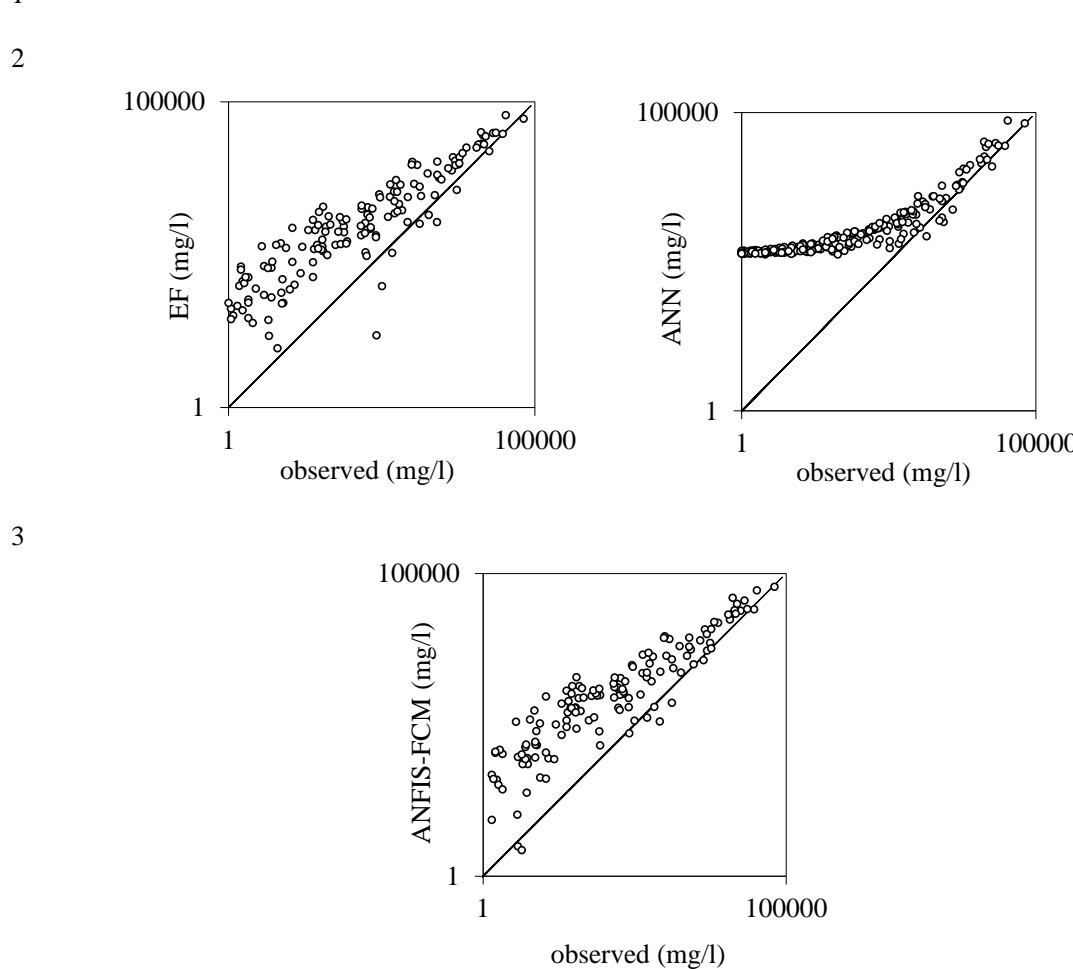

5
6          **Figure 8**



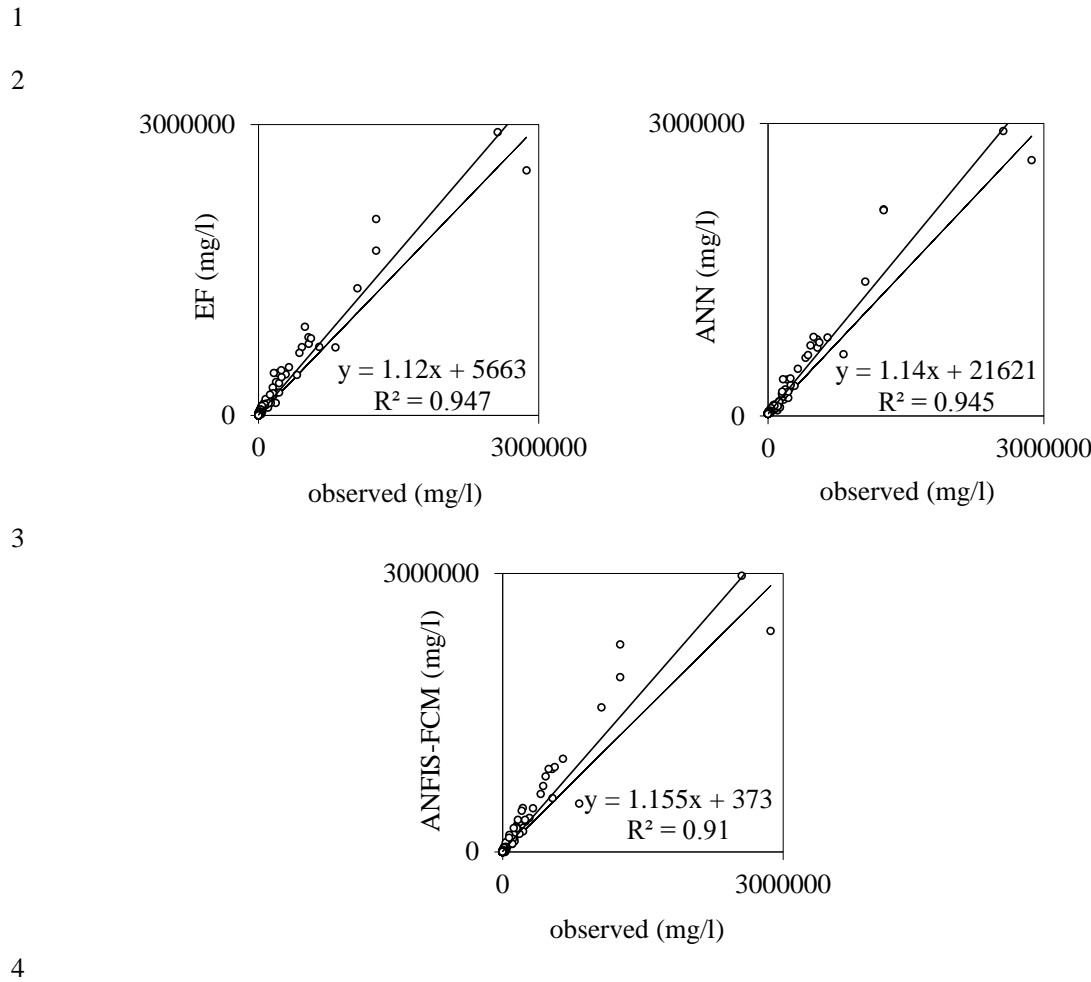

5
6
7                        **Figure 9**





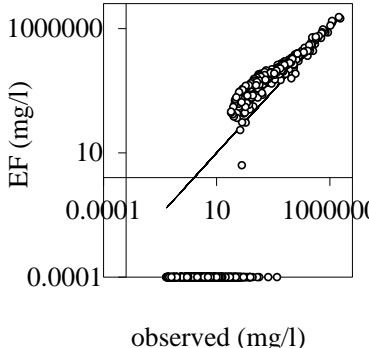
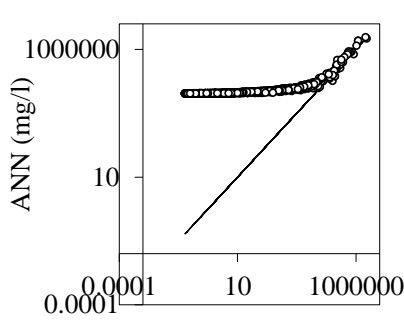

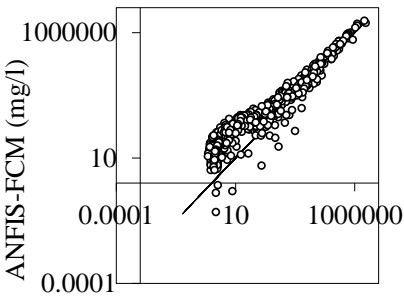

6                            **Figure 10**





**TABLES**
**Table 1.** The daily statistical parameters of data set for the stations.

| Data set | Station | Basin area (km$^2$) | Data type | $x_{mean}$ | $S_x$ | $C_v$ ($S_x/x_{mean}$) | $C_{sx}$ | $x_{max}$ | $x_{min}$ | $\frac{x_{max}}{x_{mean}}$ |
|---|---|---|---|---|---|---|---|---|---|---|
| Training | Upstream (11472150) | 1368 | Flow (m$^3$ s$^{-1}$) | 32.7 | 92.8 | 2.83 | 6.31 | 1270 | 0.05 | 38.8 |
| | | | Sediment (mg l$^{-1}$) | 2790 | 17076 | 6.12 | 10.8 | 318000 | 0 | 114 |
| | Downstream (11477000) | 8063 | Flow (m$^3$ s$^{-1}$) | 266 | 625 | 2.35 | 5.01 | 7560 | 2.07 | 28.4 |
| | | | Sediment (mg l$^{-1}$) | 60966 | 288396 | 4.73 | 8.50 | 4930000 | 0.23 | 80.9 |
| Validation | Upstream (11472150) | 1368 | Flow (m$^3$ s$^{-1}$) | 33.3 | 95.5 | 2.86 | 8.02 | 1590 | 0.08 | 47.7 |
| | | | Sediment (mg l$^{-1}$) | 2706 | 25108 | 9.28 | 19.5 | 661000 | 0 | 244 |
| | Downstream (11477000) | 8063 | Flow (m$^3$ s$^{-1}$) | 296 | 693 | 2.34 | 6.32 | 9170 | 2.32 | 31.0 |
| | | | Sediment (mg l$^{-1}$) | 46210 | 303083 | 6.55 | 14.4 | 6230000 | 0 | 135 |
| Test | Upstream (11472150) | 1368 | Flow (m$^3$ s$^{-1}$) | 12.1 | 49.1 | 4.05 | 8.05 | 680 | 0 | 56.1 |
| | | | Sediment (mg l$^{-1}$) | 561 | 4780 | 8.52 | 13.1 | 86500 | 0 | 154 |
| | Downstream (11477000) | 8063 | Flow (m$^3$ s$^{-1}$) | 131 | 408 | 3.12 | 7.11 | 5270 | 0.71 | 40.3 |
| | | | Sediment (mg l$^{-1}$) | 18432 | 143023 | 7.76 | 14.5 | 2870000 | 0.06 | 156 |

**Table 2.** The cross-corelations between disharge and SSC in the upstream and downstream
stations.

| | $Q_t$ | $Q_{t-1}$ | $Q_{t-2}$ | $Q_{t-3}$ | $Q_{t-4}$ | $Q_{t-5}$ |
|---|---|---|---|---|---|---|
| Upstream St | 0.884 | 0.545 | 0.353 | 0.353 | 0.310 | 0.275 |
| Downstream St | 0.873 | 0.524 | 0.358 | 0.358 | 0.317 | 0.259 |




**Table 3.** The traning and validation performances of the EF, ANN and ANFIS-FCM models in
suspended sediment prediction – Upstream station

| Models | Inputs | Training time (sec) | Training RMSE (mg/l) | MAE (mg/l) | $R^2$ | Validation RMSE (mg/l) | MAE (mg/l) | $R^2$ |
|---|---|---|---|---|---|---|---|---|
| EF(2,gauss,10000) | i)   Qt | 1047 | 5482 | 1378 | 0.898 | 9704 | 1604 | 0.912 |
| EF(2,gauss,1000) | ii)  Qt, Qt-1 | 144 | 4876 | 1190 | 0.919 | 9783 | 1415 | 0.908 |
| EF(3,gauss,5000) | iii) Qt, Qt-1, Qt-2 | 2442 | 4579 | 983 | 0.928 | 9973 | 1259 | 0.908 |
| EF(3,gauss,5000) | iv) Qt, Qt-1, Qt-2, Qt-3 | 5211 | 4436 | 1095 | 0.933 | 12232 | 1610 | 0.816 |
| ANN(1,1,1) | i)   Qt | 6.96 | 6175 | 2242 | 0.874 | 13097 | 2389 | 0.818 |
| ANN(2,2,1) | ii)  Qt, Qt-1 | 7.64 | 4177 | 1149 | 0.941 | 11836 | 1568 | 0.824 |
| ANN(3,2,1) | iii) Qt, Qt-1, Qt-2 | 7.78 | 4176 | 1152 | 0.941 | 11837 | 1567 | 0.838 |
| ANN(4,2,1) | iv) Qt, Qt-1, Qt-2, Qt-3 | 7.92 | 4162 | 1173 | 0.941 | 11826 | 1593 | 0.841 |
| ANFIS-FCM(8) | i)   Qt | 0.75 | 4703 | 943 | 0.924 | 20784 | 1602 | 0.315 |
| ANFIS-FCM(7) | ii)  Qt, Qt-1 | 0.31 | 3966 | 930 | 0.946 | 6390 | 1012 | 0.948 |
| ANFIS-FCM(5) | iii) Qt, Qt-1, Qt-2 | 1.14 | 4379 | 1016 | 0.934 | 9032 | 1248 | 0.912 |
| ANFIS-FCM(5) | iv) Qt, Qt-1, Qt-2, Qt-3 | 3.58 | 4319 | 1258 | 0.937 | 8524 | 1580 | 0.923 |





2 **Table 4.** The traning and validation performances of the EF, ANN and ANFIS-FCM models in
3 suspended sediment prediction – Downstream station

| Models | Inputs | Training time (sec) | Training RMSE (mg/l) | MAE (mg/l) | $R^2$ | Validation RMSE (mg/l) | MAE (mg/l) | $R^2$ |
|---|---|---|---|---|---|---|---|---|
| EF(2,gauss,20000) | i) Qt | 2211 | 85898 | 23182 | 0.912 | 164866 | 32127 | 0.777 |
| EF(5,gauss,1000) | ii) Qt, Qt-1 | 327 | 63122 | 16028 | 0.952 | 168184 | 31977 | 0.698 |
| EF(2,gauss,50000) | iii) Qt, Qt-1, Qt-2 | 10354 | 75041 | 19719 | 0.932 | 214455 | 40579 | 0.610 |
| EF(4,gauss,5000) | iv) Qt, Qt-1, Qt-2, Qt-3 | 8768 | 72437 | 19282 | 0.937 | 182543 | 34471 | 0.701 |
| ANN(1,2,1) | i) Qt | 7.56 | 93265 | 36014 | 0.898 | 160822 | 43381 | 0.734 |
| ANN(2,2,1) | ii) Qt, Qt-1 | 7.74 | 75190 | 25994 | 0.933 | 139047 | 37361 | 0.823 |
| ANN(3,1,1) | iii) Qt, Qt-1, Qt-2 | 7.46 | 89231 | 34060 | 0.907 | 168271 | 44889 | 0.708 |
| ANN(4,1,1) | iv) Qt, Qt-1, Qt-2, Qt-3 | 8.68 | 89035 | 34651 | 0.907 | 167950 | 45222 | 0.709 |
| ANFIS-FCM(2) | i) Qt | 0.45 | 87038 | 24817 | 0.909 | 157874 | 34597 | 0.773 |
| ANFIS-FCM(8) | ii) Qt, Qt-1 | 0.35 | 73337 | 16598 | 0.935 | 177645 | 26975 | 0.751 |
| ANFIS-FCM(8) | iii) Qt, Qt-1, Qt-2 | 1.53 | 80011 | 17997 | 0.923 | 202776 | 29772 | 0.712 |
| ANFIS-FCM(3) | iv) Qt, Qt-1, Qt-2, Qt-3 | 0.88 | 78597 | 20174 | 0.926 | 176095 | 34031 | 0.731 |

5



2 **Table 5.** The test performances of the optimal EF, ANN and ANFIS-FCM models in suspended
3 sediment prediction

| Models | Inputs | RMSE (mg/l) | MAE (mg/l) | $R^2$ |
|---|---|---|---|---|
| Upstream station | | | | |
| EF(2,gauss,10000) | i) Qt | 2588 | 503 | 0.892 |
| EF(2,gauss,1000) | ii) Qt, Qt-1 | 2654 | 445 | 0.931 |
| EF(3,gauss,5000) | iii) Qt, Qt-1, Qt-2 | 2583 | 413 | 0.933 |
| EF(3,gauss,5000) | iv) Qt, Qt-1, Qt-2, Qt-3 | 2685 | 476 | 0.928 |
| ANN(1,1,1) | i) Qt | 3206 | 1583 | 0.875 |
| ANN(2,2,1) | ii) Qt, Qt-1 | 2999 | 708 | 0.941 |
| ANN(3,2,1) | iii) Qt, Qt-1, Qt-2 | 2989 | 709 | 0.941 |
| ANN(4,2,1) | iv) Qt, Qt-1, Qt-2, Qt-3 | 2962 | 736 | 0.943 |
| ANFIS-FCM(8) | i) Qt | 3172 | 449 | 0.904 |
| ANFIS-FCM(7) | ii) Qt, Qt-1 | 3065 | 401 | 0.926 |
| ANFIS-FCM(5) | iii) Qt, Qt-1, Qt-2 | 2912 | 416 | 0.922 |
| ANFIS-FCM(5) | iv) Qt, Qt-1, Qt-2, Qt-3 | 3022 | 499 | 0.935 |
| Downstream station | | | | |
| EF(2,gauss,20000) | i) Qt | 47773 | 10285 | 0.929 |
| EF(5,gauss,1000) | ii) Qt, Qt-1 | 44593 | 8414 | 0.939 |
| EF(2,gauss,50000) | iii) Qt, Qt-1, Qt-2 | 42714 | 9032 | 0.947 |
| EF(4,gauss,5000) | iv) Qt, Qt-1, Qt-2, Qt-3 | 45493 | 10149 | 0.948 |
| ANN(1,2,1) | i) Qt | 53083 | 25489 | 0.927 |
| ANN(2,2,1) | ii) Qt, Qt-1 | 52215 | 18227 | 0.907 |
| ANN(3,1,1) | iii) Qt, Qt-1, Qt-2 | 50986 | 24917 | 0.944 |
| ANN(4,1,1) | iv) Qt, Qt-1, Qt-2, Qt-3 | 50491 | 25444 | 0.945 |
| ANFIS-FCM(2) | i) Qt | 50499 | 10994 | 0.921 |
| ANFIS-FCM(8) | ii) Qt, Qt-1 | 54149 | 8591 | 0.919 |
| ANFIS-FCM(8) | iii) Qt, Qt-1, Qt-2 | 45569 | 1498 | 0.940 |
| ANFIS-FCM(3) | iv) Qt, Qt-1, Qt-2, Qt-3 | 51721 | 11229 | 0.937 |





**Table 6.** The comparison of EF, ANN and ANFIS-FCM peak-estimates for the test period-
Upstream station.

| Day | Peaks > 15000 (mg/l) | EF (mg/l) | ANN (mg/l) | ANFIS-FCM (mg/l) | Relative Error | | |
|---|---|---|---|---|---|---|---|
| | | | | | EF (%) | ANN (%) | ANFIS-FCM (%) |
| 135 | 65100 | 52375 | 65890 | 60511 | -19.5 | 1.2 | -7.0 |
| 136 | 86500 | 121380 | 124244 | 108905 | 40.3 | 43.6 | 25.9 |
| 137 | 18000 | 15494 | 12534 | 24360 | -13.9 | -30.4 | 35.3 |
| 168 | 29800 | 29905 | 27751 | 25478 | 0.4 | -6.9 | -14.5 |
| 169 | 71300 | 101871 | 116166 | 120171 | 42.9 | 62.9 | 68.5 |
| 170 | 33300 | 60004 | 73166 | 52681 | 80.2 | 120 | 58.2 |
| 172 | 20800 | 30555 | 30703 | 35895 | 46.9 | 47.6 | 72.6 |
| 175 | 15700 | 27199 | 29803 | 31601 | 73.2 | 89.8 | 101 |
| 176 | 59800 | 108508 | 116646 | 120463 | 81.5 | 95.1 | 101 |
| 514 | 23000 | 30997 | 27956 | 25787 | 34.8 | 21.5 | 12.1 |
| | | Total (Absolute) = | | | **434** | **519** | **497** |




1  **Table 7.**The comparison of EF, ANN and ANFIS-FCM peak-estimates for the test period-
2  Downstream station.

| Day | Peaks > 500000 (mg/l) | EF (mg/l) | ANN (mg/l) | ANFIS-FCM (mg/l) | Relative Error | | |
|---|---|---|---|---|---|---|---|
| | | | | | EF (%) | ANN (%) | ANFIS-FCM (%) |
| 132 | 538000 | 737820 | 700519 | 579618 | 37.1 | 30.2 | 7.7 |
| 133 | 535000 | 806291 | 783917 | 892406 | 50.7 | 46.5 | 66.8 |
| 135 | 822000 | 702169 | 631613 | 520127 | -14.6 | -23.2 | -36.7 |
| 136 | 2870000 | 2526702 | 2624083 | 2381486 | -12.0 | -8.6 | -17.0 |
| 137 | 649000 | 705882 | 802897 | 1004263 | 8.8 | 23.7 | 54.7 |
| 143 | 560000 | 793386 | 752534 | 915272 | 41.7 | 34.4 | 63.4 |
| 169 | 2560000 | 2920197 | 2922853 | 2976379 | 14.1 | 14.2 | 16.3 |
| 170 | 1260000 | 1699105 | 2116979 | 2235129 | 34.8 | 68.0 | 77.4 |
| | | | | Total (Absolute) = | **298** | **346** | **436** |

