# Peer review of "Abstract: This paper proposes the application"

_Hydrology and Earth System Sciences, 2016_

## Referee Comment (RC1) · Anonymous Referee #1 · 21 Jun 2016

Thank you for inviting me to review the above mentioned MS.

The MS is within the scope of the journal.

The MS does describe an application of an artificial intelligence methodology to the problem of prediction of daily-averaged suspended sediment concentrations (SSCs).

The used simulation methodologies are valid, but the results of the simulations have not been described and analysed in an appropriate way.

The literature references are numerous, but not all of them are analysed in the same consistent way; therefore it does look like some of the references can be removed without any loss to the content of the MS. There are a considerable/excessive number of self-references, which may need to be reduced to a necessary minimum.

[Figure]

The presentation quality in terms of the result analysis and discussion needs to be worked on.

More specific remarks are as follows.

The repetitions should be removed from the title and the abstract.

The literature review needs to be better structured, consistently with different approaches to be used in the MS.

To avoid any confusion, the used simulation approaches should be described in a similar order, consistent with the literature review and result analysis. For instance, GAs are described in detail, but it might not be clear to a reader why there are there in the first place, while any presentation of artificial neural networks (ANNs) is missing completely.

The case study section contains a description of a 1964 flood but this period is not covered by the simulations; this better be removed or clearly explained how the event is relevant to the study.

An analysis and discussion of the statistical parameters from Table 1 and explanations how those may affect the further simulations are missing.

The application and results section needs to be structured in accord to different parameters being described in there. The tables and figures mentioned here have not been properly analysed and discussed. What is "clear" to the author has not been made clear to the reader, and contradicts the presented results.

For instance, the author states: "The log-scaled scatterplots of the optimal models are compared in Figure 10 for the test period. The EF model seems to have better accuracy in estimating average and low SSC values than the ANN and ANFIS-FCM models." The scaling and labelling of the axes in the figure makes it very difficult to analyse, but conclusions which may be drawn even from a superficial analysis are that the used EF underestimates SSCs up to 100,000s, while the ANNs overestimate SSCs

up to 100,000. The ANFIS-FCM appears to be performing the most consistently for the case.

Such inconsistencies make all the conclusions questionable.

---

## Referee Comment (RC2) · Anonymous Referee #2 · 14 Jul 2016

In this manuscripts, the evolutionary fuzzy (EF) approach was applied for prediction of daily suspended sediment concentration (SSC). The EF was improved by the combination of two methods, fuzzy logic and genetic algorithm. The accuracy of EF models is compared with those of the artificial neural network (ANN) and adaptive neuro-fuzzy inference system with fuzzy c-means clustering (ANFIS-FCM). The daily streamflow and suspended sediment data collected from two stations on the Eel River in California, United States are used in the study. Comparison of the optimal EF, ANN and ANFIS-FCM models in estimating peak and total suspended sediments revealed that the EF model provided better accuracy than the ANN and ANFIS-FCM. The study technically sounds and is generally well written. However, there are some points that should be considered in the revised paper before its publication. The literature section should be re-organized by removing some of excessive references (e.g., self-citations).

[Figure]

Dates of the references are not new and do not seem to be up to date. Some recently published papers should be added in to the literature. The methods section needs extension. The ANN and ANFIS are missing in the methods section and they should be briefly explained. Which type of fuzzy system did you use in EF method? This should be clearly provided. Consequent parameters and rules of the optimal EF and ANFIS models should be provided. How the author find optimal hidden node of the ANN models? This can be explained by giving error variation with respect to hidden node number. The membership functions may be provided for the optimal models. The results obtained from the study may be compared with previous ones related to same topic. Future studies section may be added at the end of the Conclusions section.